# Biocompatible Triple-Helical Recombinant Collagen Dressings for Accelerated Wound Healing in Microneedle-Injured and Photodamaged Skin

**Caihong Fu** [1,2,3] , **Shuangni Shi** [1,2,3], **Nannan Wei** [1,2,3], **Yirui Fan** [1,2,3], **Hong Gu** [4], **Peng Liu** [2,3,*] **and Jianxi Xiao** [1,2,3,*]

1   State Key Laboratory of Applied Organic Chemistry, College of Chemistry and Chemical Engineering, Lanzhou University, Lanzhou 730000, China
2   Gansu Engineering Research Center of Medical Collagen, Lanzhou 730000, China
3   Joint Research Center of Collagen of Lanzhou University China National Biotec Group, Lanzhou Biotechnology Development Co., Ltd., Lanzhou 730000, China
4   School of Basic Medical Sciences, Lanzhou University, Lanzhou 730000, China
*   Correspondence: pengliu@vacmic.com (P.L.); xiaojx@lzu.edu.cn (J.X.)

**Abstract:** Skin rejuvenation procedures such as microneedling and laser resurfacing have gained global popularity in medical cosmetology, leading to acute skin wounds with persistent pain, erythema, and edema. A variety of dressings have been explored to repair these postoperative skin injuries; however, their inadequate biocompatibility and bioactivity may raise concerns about undesirable efficacy and complications. Herein, we developed biocompatible and nonirritating triple-helical recombinant collagen (THRC) dressings for accelerated healing of microneedle-injured and photodamaged acute skin wounds. Circular dichroism (CD) measurements of THRC from various batches exhibited triple-helical structure characteristics of collagen. Cell experiments using L929 fibroblasts revealed that THRC dressings possess superior biocompatibility and bioactivity, significantly elevating the proliferation and adhesion of fibroblasts. In vivo, skin irritation tests of New Zealand rabbits demonstrated that the THRC dressings are gentle, safe, and non-irritating. Histological analysis of the animal model studies in photodamaged skin wounds using H&E and Masson's trichrome staining revealed that 4 days of treatment with the THRC dressings effectively healed the damaged dermis by accelerating re-epithelialization and enhancing collagen deposition. In vivo studies of microneedle-injured rat defects showed that THRC dressings of varying concentrations exhibit the same rapid epithelialization rates at 48 h as commercial bovine collagen dressings. The highly biocompatible and bioactive recombinant collagen dressings may provide an advanced treatment of acute skin wounds, indicating attractive applications in postoperative care of facial rejuvenation.

**Keywords:** recombinant collagen dressing; skin rejuvenation; wound healing; microneedling; photodamaged skin

## 1. Introduction

Skin rejuvenation procedures including microneedling and laser resurfacing have gained popularity worldwide for improving facial appearance [1,2]. Microneedling physically destroys the epidermis and papillary dermis, thus stimulating collagen regeneration against skin aging, while laser procedures can induce multiple skin ablation by activating the wound healing process to repair scars and other complications [3,4]. Microneedling and laser procedures are utilized to intentionally injure skin for cosmetic purposes, leading to wounds with persistent pain, postoperative erythema, and acute inflammation [5,6]. Wound care dressings are increasingly necessary to shorten the recovery time of acute symptoms and prevent skin pigmentation and scar formation, overall improving facial rejuvenation outcomes [7].

Various wound care dressings have been developed for skin rejuvenation post-procedure treatments. Petrolatum-based ointments have been commonly used for post-treatments, which are too occlusive to promote wound maceration and may cause contact dermatitis [8,9]. Silicone-based gels are semi-occlusive dressings that prevent maceration, but they may be insufficient in accelerating skin healing [10–13]. A formulation containing tri- and hexapeptides has been applied to reduce skin redness after radiofrequency (RF) microneedling [14,15]. Other wound care dressings have also been explored for postoperative management, achieving differential degrees of success [16,17]. Unfortunately, these currently available wound dressings display limitations, such as poor bioactivity and ineffectiveness in reducing pain, accelerating epithelialization, or preventing hyperpigmentation in severely photodamaged skin [18]. Better skin wound dressings with high bioactivity for topical postoperative management remain very demanding.

Collagen dressings with high bioactivity and biocompatibility have been extensively applied to postoperative wounds [19]. Collagen is the main structural component of the dermis and plays a key role in wound healing phases, which provides supporting scaffolds for wounds and further influences cell behaviors by regulating matrix deposition [20]. Collagen wound care dressings for post-treatments with the advantages of maintaining a bioactive microenvironment for cell growth, promoting matrix remodeling, and relieving pain postoperatively have been investigated [21]. A collagen membrane was constructed to treat surgical defects, improving wound healing by reducing pain and accelerating epithelialization [22]. A lyophilized bovine collagen matrix was developed for postoperative wound management, providing faster wound healing rates than the traditional methods [23]. However, the shortages of animal-derived collagen such as allergic reactions and the risks of disease transfer need to be noticed and overcame [24]. Novel recombinant collagen with low immunogenicity and disease transmission provides an attractive alternative to the usages of animal-derived collagen dressings [25].

Recombinant collagen with high purity, low immunogenicity, and batch-to-batch consistency is becoming increasingly popular in wound dressings. To address the issues with animal-derived collagen and to provide substantial sources of collagen, recombinant collagens have been investigated using various expression systems (e.g., bacteria, yeast, and insect cells) [26–28]. As native-like collagen, recombinant collagen should possess biological activity by acting as signal biomolecules to promote cell proliferation and migration, which requires cell-binding sites and rod-like triple-helical structures [29,30]. The stability of collagen triple helices is significantly related to its high melting temperature, which is critical for bioactive collagen wound care dressings to maintain stable folding beyond room temperature [31,32]. Due to the complexity of the collagen structure, the production of triple-helical recombinant collagen wound dressings with bioactivity and high stability remains a major challenge.

Here, we construct a triple-helical recombinant collagen dressing with high bioactivity and no skin irritation, which shows accelerated healing efficacy in microneedle-injured and photodamaged acute wounds. The genetically engineered collagen is produced by an Escherichia coli (*E. coli*) expression system with a perfect triple-helix structure and a high melting temperature of 34 °C. The triple-helical recombinant collagen (THRC) is utilized to fabricate the recombinant collagen dressing (THRC dressing). THRC dressings with high biocompatibility and bioactivity promoting fibroblast proliferation and adhesion have been applied for the post-treatments of microneedle-injured and photodamaged wounds, which display excellent healing efficacies by accelerating re-epithelialization and enhancing collagen deposition. THRC dressings with ultimate safety and bioactivity are promising treatments postoperatively in facial rejuvenation.

## 2. Materials and Methods

### 2.1. Preparation of THRC and THRC Dressings

Protein THRC was expressed by *Escherichia coli (E. coli) BL21* strain using plasmid pColdIII-THRC following previously reported protocols [33]. The cells were cultured in 50

mL of LB medium with 100 mg/L ampicillin overnight at 37 °C. The cells were transferred to 1 L of LB medium and incubated at 37 °C. When the $OD_{600}$ reached 0.5–2.0, the temperature decreased from 37 °C to 25 °C, and 1 mM isopropyl beta-D-thiogalactopyranoside (IPTG) was added to induce protein expression. After 8–36 h of incubation, the cells were collected and resuspended in binding buffer (20 mM sodium phosphate buffer, pH 7.4, 500 mM NaCl, and 20 mM imidazole). The cells were lysed by a homogenizer, and the supernatant fraction was harvested. Raw proteins were purified using a Ni-NTA Sepharose column with elution buffer (20 mM sodium phosphate buffer pH 7.4, 500 mM NaCl, 500 mM imidazole). Collagen protein THRC was obtained after trypsin digestion of the purified protein to remove the folding domain as previously described [34]. The degraded fragments were removed by dialysis using 50 mM PBS buffer (pH 7.4). The purified THRC was confirmed by SDS-PAGE and lyophilized for future use.

THRC solutions (5 mg/mL) were diluted into different concentrations (0.1, 0.5, and 1.0 mg/mL), and a few medical preservatives (phenoxyethanol, etc.) were added and gently homogenized into uniform mixtures. The dressing mixtures were filtered through a 0.22 μm membrane and transferred to the aluminum bags containing nonwoven fabric irradiated by Co-60 in advance. Every aluminum bag of 28 mL dressing mixture was sealed and stored at 4 °C. All ingredients used in the preparation of THRC and THRC dressings were of medical grade, and all the procedures were carried out in a Grade C sterile workshop. Recombinant collagen dressings THRC-1 (0.1 mg/mL THRC), THRC-2 (0.5 mg/mL THRC), and THRC-3 (1.0 mg/mL THRC) were obtained.

### 2.2. Circular Dichroism Characterization of THRC

Circular dichroism (CD) is an absorption spectroscopy utilizing circularly polarized light to investigate structural aspects of optically active molecules, and it has been widely applied to evaluate the secondary structure and folding of proteins [35]. CD spectra were acquired on a CD spectrometer (JASCO J1500) equipped with a temperature controller. THRC samples with concentrations of 0.5 mg/mL and 1.0 mg/mL were prepared in phosphate buffer (20 mM, pH 7.0). The samples were equilibrated for at least 24 h at 4 °C before the CD measurements. Cells with a path length of 1 mm were used. Wavelength scans were carried out using three samples of 0.5 mg/mL THRC from 190 to 300 nm with a 1.0 nm increment per step and a 1.0 s averaging time. Thermal unfolding curves were measured utilizing three samples of 1.0 mg/mL THRC by monitoring the amplitude of the characteristic CD band at 221 nm, while the temperature was increased by 1 °C/min from 4 °C to 20 °C, 0.3 °C/min from 20 °C to 50 °C, and 1 °C/min from 50 °C to 60 °C with an equilibration time of 1 min at each temperature. The melting temperature (Tm) was estimated from the first derivative of the thermal unfolding curves.

### 2.3. Cytotoxicity of THRC Dressing

The L929 cell line was purchased from the Microbial Culture Preservation Center of the Chinese Academy of Sciences (Beijing, China). The biocompatibility of THRC dressings was investigated using L929 fibroblasts. All THRC dressings (<0.1 mg/mL THRC) used in cell experiments were diluted with THRC-1 dressing. L929 fibroblasts (5000 cells/well) were added to a 96-well plate and incubated for 24 h. DMEM was used as a blank control, and THRC dressings at various concentrations of THRC (0.001, 0.005, 0.01, 0.05, and 0.1 mg/mL) were added to six wells and cocultured for 24 h. Subsequently, 10 μL of CCK-8 was added and incubated in dark for 4 h at 37 °C. The $OD_{450}$ of every well was measured using a Tecan Infinite F200/M200 multifunction microplate reader (Tecan, Männedorf, Switzerland). The cell viability of THRC dressing at different concentrations was calculated by comparison with the blank. The blank control was set as 100% of cell viability.

### 2.4. Cell Proliferation of THRC Dressing

Cell proliferation experiments were performed to further evaluate the biocompatibility of THRC dressings. L929 fibroblasts (5000 cells/well) were added to a 96-well plate and incubated for 24 h. DMEM was used as a blank control, and 0.1 mg/mL THRC dressing in DMEM medium was added to 6 wells. L929 cells were cultured at 37 °C for 24 h, 48 h, 72 h, and 96 h, respectively. At every time interval, 10 μL of CCK-8 was added and incubated in the dark for 4 h at 37 °C. Subsequently, the $OD_{450}$ of every well was measured. The relative growth rates of THRC dressing on different days were calculated by comparing with the blank. The blank control was set as 100% of the growth rate.

### 2.5. Cell Adhesion of THRC Dressing

Cell adhesion assays were performed to study the biological activity of THRC dressings. A 24-well cell culture plate with no TC treatment was used. Then, 0.1 mg/mL THRC dressing was added to three wells, while heat-denatured 1% bovine serum albumin (BSA) was added as a non-adhesive control. The plate was air-dried overnight, and 500 μL of L929 cell suspension in DMEM ($3 \times 10^5$ cells/mL) was added. The plate was incubated at 37 °C for 5 h. The images of cells were acquired using a Leica fluorescence microscope (Leica Microsystems Inc., Wetzlar, Germany).

### 2.6. Skin Irritation Test of THRC Dressing on Rabbits

All animal experiments were performed with protocols approved by the ethics committee of the College of Chemistry and Chemical Engineering at Lanzhou University (No. G09, 20220711). Rabbits, SD rats, and mice were purchased from Lanzhou Veterinary Research Institute (Lanzhou, China). All animals were fed with standard laboratory diets referring to the Code of Practice for the Housing and Care of Animals Used in Scientific Procedures. All animal research complied with the commonly accepted "3Rs".

A skin irritation test was performed to evaluate the tolerance of rabbit skin to THRC dressings based on GB/T 16886.10–2005 (China) and the United Nations Globally Harmonized System of Classification and Labeling of Chemicals (GHS). All rabbits weighed $2.0 \pm 0.2$ kg and were between 3 and 4 months of age. Five New Zealand white rabbits (two males and three nonpregnant females) with healthy skin were used. A female rabbit was used for pre-experiment to test the feasibility of experimental procedures. The pre-experiment was successful; thus, a formal skin irritation test was conducted with another four rabbits. Four rabbits were fed in a single cage and adapted for 3 days. Then, 24 h before the experiment, fur from the backs of the four rabbits was removed and clipped into 10 cm × 15 cm areas with no damage to the epidermis. THRC-3 dressings (2.5 cm × 2.5 cm) were applied in the test areas gently, while the untreated areas were controlled. After 4 h, the dressings were taken off, and the residual liquid was removed with warm water. After removing the test dressings at different time intervals (1 h, 24 h, 48 h, and 72 h), the skin of the test and control areas was observed under natural light and quantitatively scored for erythema and edema according to the scoring standards for erythema and edema formation: no erythema, 0 points; very slight erythema (barely visible), 1 point; clear erythema, 2 points; moderate erythema, 3 points; severe erythema (purple-red) to eschar formation, 4 points; no edema, 0 points; very slight edema (barely visible), 1 point; clear edema (swelling, not beyond the edge of the area), 2 points; moderate edema (about 1 mm swelling), 3 points; severe edema (the swelling exceeds 1 mm and exceeds the contact area), 4 points. According to the GHS system, the primary dermal irritation indices (PDIIs) were calculated according to the following formula [20,21].

$$\text{PDII} = \frac{[\text{Sum erythema (all time points)} + \text{Sum edema (all time points)}]}{\text{number of intervals} \times \text{number of animals}}. \tag{1}$$

### 2.7. Animal Experiment of THRC Dressing in Photodamaged Skin Healing

The photodamaged skin model was established by excessive UV exposure to study the repair efficiency of THRC dressing for acute photodamaged wounds. The 30 female mice used in this experiment were between 6 and 8 weeks of age, weighing $29 \pm 2$ g. Before the test, all mice were depilated on the back with a 2 cm $\times$ 4 cm area. Firstly, 30 mice were randomly divided into three groups with 10 mice in each group: the normal group, the untreated group, and the THRC dressing group. Ten mice in the normal group acted as healthy skin controls without UV irradiation. Secondly, all mice in untreated and THRC dressing groups were irradiated with UVA (320–440 nm) and UVB (280–320 nm) lamps (Nanjing Huaqiang Electronics Co., Ltd., Nanjing, China) of 100 mJ/cm$^2$ to induce acute skin inflammation [22,23]. Ten UV-mediated photodamaged mice in the untreated group were used as negative controls without any treatments. Immediately post UV radiation on day 0, THRC-3 dressing was applied to the acute wounds of mice in the THRC dressing group. Daily treatment of 20 min was applied on day 1 and day 4 in the THRC dressing group. For each group, five mice were sacrificed on day 2 and day 4, respectively. The wound sites of mice skin were harvested and paraffin-sectioned. H&E staining and Masson's trichrome staining were then performed. The stained sections were imaged on a Leica DM4000B metallurgical upright microscope (Leica Microsystems Inc., Wetzlar, Germany).

### 2.8. Animal Experiment of THRC Dressing in Microneedle-Injured Skin Healing

The microneedle-injured skin model was established to evaluate the healing efficacy of THRC dressings compared with commercial bovine collagen dressing Traucr$^{TM}$ (Guangzhou Trauer Biotechnology Co., Ltd. Guangzhou, China). A total of 36 SD rats were used in the experiment with half male and half nonpregnant females. All rats were in a body weight range of 180–200 g and between 2 and 3 months of age. The rats were raised adaptively for 3 days. Before the test, the dorsal hair of the rats was removed from the test site. Firstly, 36 rats were randomly divided into six groups with three males and three females in each group: the blank control group, the negative control group, the Traucr$^{TM}$ group, the THRC-3 group, the THRC-2 group, and the THRC-1 group. Secondly, the depilated skin of rats was repeatedly pasted with adhesive tape. After obvious redness, a microneedle (0.5 mm length) was then applied to destroy the epidermis and injure the dermis.

Subsequently, the dressing treatments were applied to six groups as follows: blank control group, 2.5 cm $\times$ 2.5 cm dry non-woven fabric was applied to wound sites; negative control group, 2.5 cm $\times$ 2.5 cm non-woven fabric soaked in 0.9% saline was applied to wound sites; Traucr$^{TM}$ group, the commercial bovine collagen dressing was cut into the size (2.5 cm $\times$ 2.5 cm) and applied to act as a positive control; THRC-3 group, THRC-2 group, and THRC-1 group, THRC-3, THRC-2, and THRC-1 dressings were cut into the size (2.5 cm $\times$ 2.5 cm) and applied to the wound areas, respectively. All rats were treated immediately post microneedling (0 h) and at different time intervals (12 h, 24 h, and 48 h). The bandages were used for fixation during the application. After 20 min, the bandages and the dressings were removed. At 0 h, 24 h, and 48 h, a female rat and a male rat in each group were randomly selected for morphological observation and sacrificed. The wound sites were sampled and then paraffin-sectioned. Subsequently, hematoxylin and eosin (H&E) staining was performed. The images of H&E-stained sections were acquired on a Leica DM4000B metallurgical upright microscope (Leica Microsystems Inc., Wetzlar, Germany).

### 2.9. Statistics

Student's *t*-test was utilized to compare the two independent groups for the quantitative analysis of cell proliferation and collagen volume fraction. Differences were considered to be statistically significant when the *p*-value was <0.05.

## 3. Results

### 3.1. Structure Characterization of THRC

Circular dichroism (CD) was performed to determine the triple-helix structure and the melting temperature (Tm) of THRC. THRC samples from three different batches (sample 1, sample 2, and sample 3) were examined. A positive peak near 221 nm and a negative peak near 198 nm were indicative of the triple-helix structure of collagen [24,25]. CD spectra of sample 1, sample 2, and sample 3 all showed positive peaks around 221 nm and negative peaks around 198 nm, indicating the formation of a distinct triple-helical structure of THRC (Figure 1A). The amplitude of the CD absorption at 221 nm was monitored from 4 °C to 60 °C to obtain the thermal transition curve (Figure 1(Ba)), while Tm was calculated from the first derivative of the thermal unfolding curves (Figure 1(Bb)). The melting temperature of THRC was estimated as 34 °C in all three samples, indicating the high stability of the triple-helix structure of THRC.

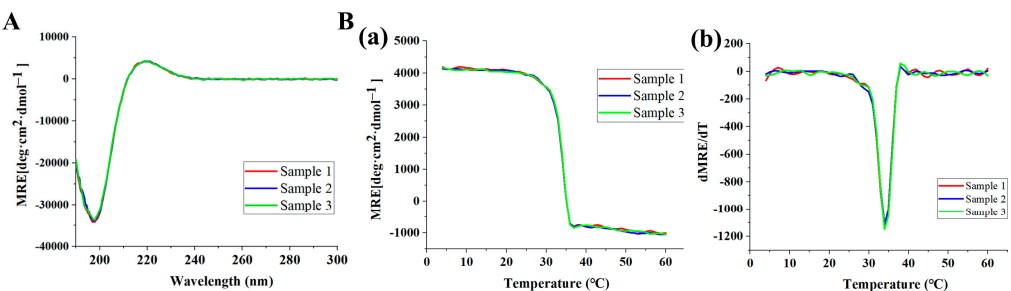

**Figure 1.** CD spectrum of THRC samples from different batches (sample 1, sample 2, and sample 3). (**A**) CD thermal transition curves; (**B**) CD thermal unfolding curves (**a**) and the first derivative of the thermal unfolding curves (**b**).

### 3.2. In Vitro Biocompatibility and Bioactivity of THRC Dressing

The biocompatibility of THRC dressing was evaluated by examining the viability of L929 fibroblasts using a CCK-8 kit in cytotoxicity and cell proliferation experiments. Compared with the blank control (DMEM), THRC dressing at different concentrations of THRC (0.001–0.1 mg/mL) showed greater cell viabilities (>100%), indicating that THRC dressing was nontoxic to skin fibroblasts (Figure 2A). Furthermore, compared with the DMEM blank (100%), the relative growth rates of cells cultured in 0.1 mg/ mL THRC dressing solution were all statistically higher than 100% on different time intervals (day 1, day 2, day 3, and day 4), indicating that THRC dressing significantly promoted the proliferation of skin fibroblasts (Figure 2B). The cytotoxicity and cell proliferation results demonstrated the high biocompatibility of THRC dressing.

The bioactivity of THRC was studied by an in vitro cell adhesion experiment (Figure 2C). L929 fibroblasts were cultured on the THRC dressing (0.1 mg/mL THRC) and 1% heat-denatured bovine serum albumin (BSA) substrates. After 5 h for attachments, the fibroblasts growing on the BSA surface showed a uniform spherical shape (Figure 2(Ca)), while the fibroblasts on the THRC dressing substrate displayed a well-spread pattern (Figure 2(Cb)). The result indicated that THRC dressing provided a bioactive scaffold for skin fibroblasts to adhere.

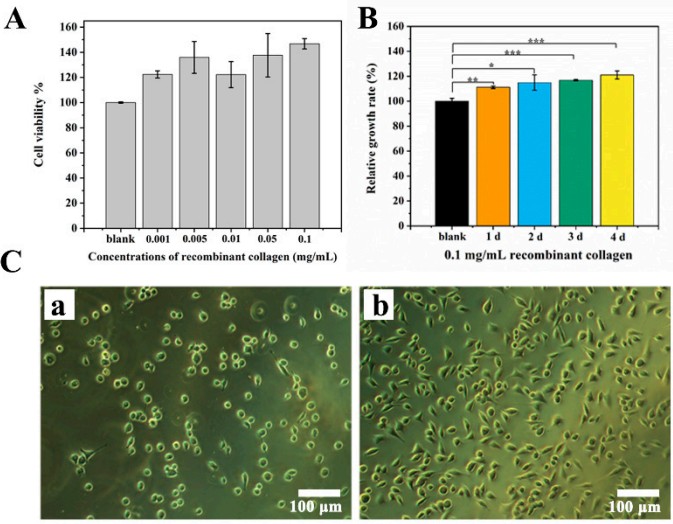

**Figure 2.** In vitro biocompatibility and bioactivity of THRC dressing. (**A**) Cytotoxicity to L929 fibroblasts of THRC dressing at various concentrations of THRC (0.001, 0.005, 0.01, 0.05, and 0.1 mg/mL). (**B**) The relative growth rates of L929 fibroblasts cultured in 0.1 mg/mL THRC dressing solution on days 1, 2, 3, and 4. (**C**) Cell adhesion images on (**a**) 1% heat-denatured BSA and (**b**) 0.1 mg/mL THRC dressing surfaces. * $p < 0.05$, ** $p < 0.01$, *** $p < 0.001$, ns $> 0.05$, $n = 6$.

### 3.3. Skin Irritation Evaluation of THRC Dressing on Rabbits

The skin irritation assessment was performed by an in vivo rabbit test to evaluate the safety of THRC dressing applied to the skin (Figure 3). The shaved rabbit skin was first divided into four parts (Figure 3(Aa)). THRC dressings (2.5 cm × 2.5 cm) were then applied to two test sites, leaving the other two sites as controls (Figure 3(Ab)). Subsequently, the bandages were applied to fix THRC dressings (Figure 3(Ac)). After 4 h of treatment, the bandages and the dressings were removed (Figure 3(Ad)) and quantitatively scored at 1 h, 24 h, 48 h, and 72 h for erythema and edema according to the standard scoring. The images of four test rabbits (Figure 3B) showed that no erythema, edema, or other complications were observed on the THRC-treated sites compared with the control areas at any time intervals (1 h, 24 h, 48 h, and 72 h). The scores of skin edema and erythema of the control and test areas were both "0" in each rabbit at any time point (Table 1). The primary dermal irritation index (PDII) was calculated as 0 according to the scores. According to the PDII irritation category, PDII = 0 represented green, indicating that the skin irritation caused by the THRC dressing (1.0 mg/mL recombinant collagen) was negligible [21]. The results indicated that THRC dressing exhibited no skin irritation.

**Table 1.** Scores of edema and erythema of control (Con) and test (Test) sites of four rabbits at various time intervals.

| Skin Reactions | 1 h | | | | 24 h | | | | 48 h | | | | 72 h | | | |
|---|---|---|---|---|---|---|---|---|---|---|---|---|---|---|---|---|
| | Edema | | Erythema | | Edema | | Erythema | | Edema | | Erythema | | Edema | | Erythema | |
| Skin sites | Con | Test | Con | Test | Con | Test | Con | Test | Con | Test | Con | Test | Con | Test | Con | Test |
| Female 1 | 0 | 0 | 0 | 0 | 0 | 0 | 0 | 0 | 0 | 0 | 0 | 0 | 0 | 0 | 0 | 0 |
| Female 2 | 0 | 0 | 0 | 0 | 0 | 0 | 0 | 0 | 0 | 0 | 0 | 0 | 0 | 0 | 0 | 0 |
| Male 1 | 0 | 0 | 0 | 0 | 0 | 0 | 0 | 0 | 0 | 0 | 0 | 0 | 0 | 0 | 0 | 0 |
| Male 2 | 0 | 0 | 0 | 0 | 0 | 0 | 0 | 0 | 0 | 0 | 0 | 0 | 0 | 0 | 0 | 0 |

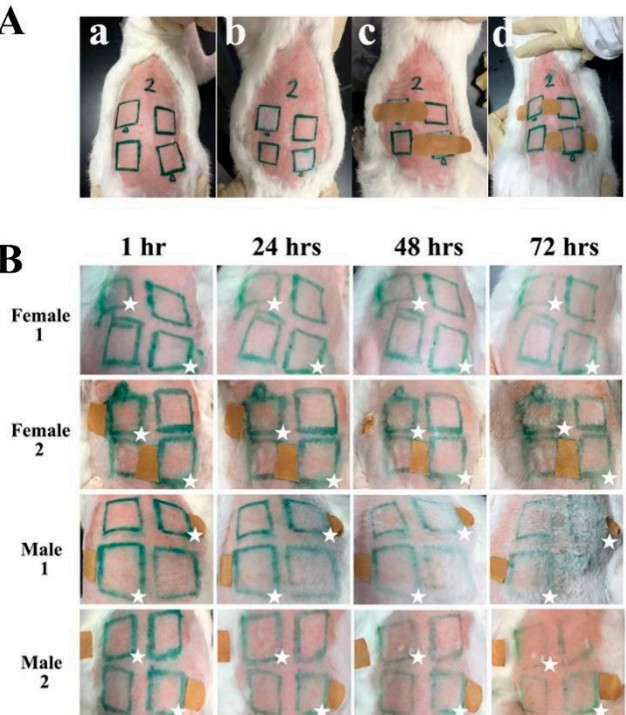

**Figure 3.** Skin irritation test of THRC dressing on rabbits. (**A**) Photos of the experimental process. The shaved rabbit skin was divided into different areas (**a**), and THRC dressings were applied on two test sites, leaving the other two sites as a control (**b**). With bandage fixation, treatments of THRC dressing were performed for 4 h (**c**), and then the dressings were removed and quantitatively scored for erythema and edema (**d**). (**B**) The images of four rabbits (Female 1, Female 2, Male 1, and Male 2) in the skin irritation test at different time intervals (1 h, 24 h, 48 h, and 72 h). The test areas are marked with ☆.

### 3.4. The Accelerated Healing Effects of THRC Dressing on Photodamaged Mouse Model

The photodamaged mouse skin model was established to evaluate the healing capacity of THRC dressing. Severe postoperative erythema and edema were observed on the shaved skin of the mice after excessive exposure to UVA and UVB (Figure 4A), indicating that photodamaged wounds with acute inflammation were formed as previously described [22,23]. The postoperative lesions were treated with THRC dressings daily and almost completely healed with no edema, little erythema, no pigmentation, and no scarring on day 4 (Figure 4B), indicating the accelerated healing effects of THRC dressing to improve cosmetic appearance.

H&E staining was performed to investigate the effectiveness of THRC dressing from a histopathological perspective (Figure 5A). Normal skin showed a complete epidermis with full layer structures and healthy dermis with no inflammation. However, the photodamaged skin displayed a destroyed epidermis with cleavages and dermal inflammation on day 2 in the untreated group, while a better epidermis and less dermal cellular infiltration after 2 days of treatments were observed in the THRC dressing group. Moreover, THRC-treated skin tissue resembled normal skin on day 4 without complications, while the epidermis remained corroded, and dermal edema persisted on the untreated wounds. The results indicated that THRC dressing healed photodamaged wounds by accelerating epithelization and reducing dermal edema.

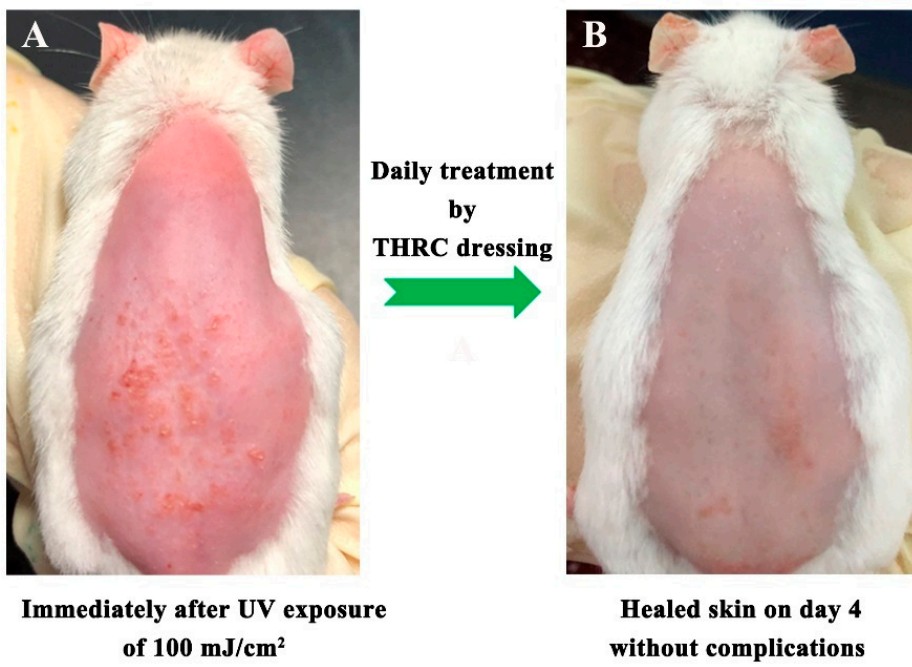

**Figure 4.** Photographs of the rat skin immediately after UV irradiation of 100 mJ/cm$^2$ with severe edema and erythema (**A**) and the healed skin after 4 days of treatment with THRC dressing without edema, erythema, and other complications (**B**).

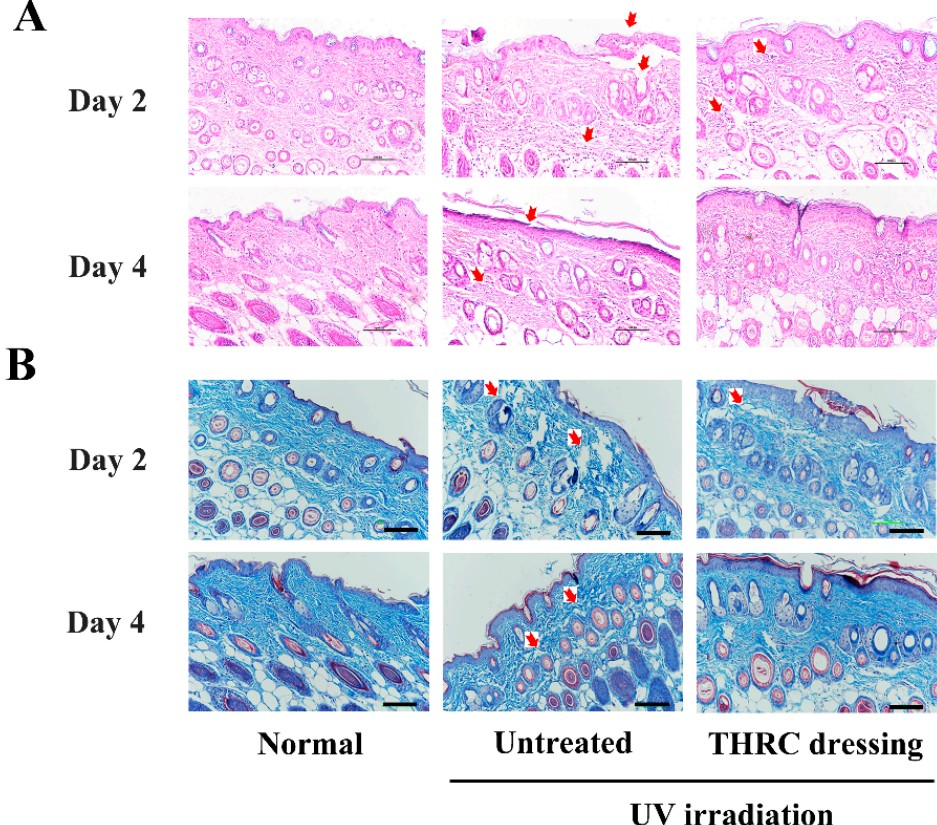

**Figure 5.** Histological analysis of normal skin and photodamaged mice acute wounds. (**A**) H&E staining and (**B**) Masson staining of skin tissues of the normal, the untreated, and the THRC dressing groups. The injuries in the epidermis and the dermis were noted by red arrows. Scale bar: 100 μm.

Masson's trichrome staining was conducted on the skin tissues to further explore the matrix remodeling efficacy of THRC dressing (Figure 5B). Well-arranged collagen fibers were observed in the normal dermis, while several collagen fibers were broken into debris on both day 2 and day 4 in the untreated group due to severe UV radiation [36]. Meanwhile, fewer broken collagen fibers were observed on day 2, and little collagen debris was found on day 4 in the THRC dressing group, indicating the accelerated regeneration of collagen fibers by THRC dressing. Furthermore, the relative content of collagen fibers in normal skin was 66.75%, which was significantly higher than the untreated tissue (46%, $p < 0.05$), indicating the enhanced degradation of collagen in photodamaged defects (Table 2). Notably, the photodamaged wounds after 4 days of treatments by THRC dressing exhibited a high collagen volume fraction (70.5%, $p > 0.05$) similar to that of the normal skin, indicating that the excessive collagen degradation due to severe UV injuries was remedied by THRC dressing. The results indicated that THRC dressing promoted collagen deposition to accelerate photodamaged wound healing.

**Table 2.** Collagen fraction volume of skin tissues in the normal, untreated, and THRC dressing groups.

| Groups | Collagen Fraction Volume (%) (Mean $\pm$ SD) | *p*-Value |
|---|---|---|
| Normal | 66.75 $\pm$ 2.56 | – |
| Untreated | 46.00 $\pm$ 1.86 | 0.000337 |
| THRC dressing | 70.50 $\pm$ 4.30 | 0.272365 |

*3.5. Evaluation of the Effectiveness of THRC Dressings with Various Concentrations on the Microneedle-Injured Rat Model*

The experiment of microneedle-injured rat post-treatments was carried out to evaluate the effectiveness of THRC dressings for postoperative wound management (Figure 6). A 0.5 mm microneedle was used to physically injure the shaved skin of rats, and acute wounds with severe erythema formed immediately after microneedling (Figure 6(Aa)) [14]. Different dressings (Table 3) were applied to the wound sites immediately postoperatively with bandage fixation for 20 min (Figure 6(Ab)). The post-treatment was applied subsequently at different time intervals (12 h, 24 h, and 48 h).

**Table 3.** Compositions of the control and test dressings on a microneedle-injured rat model.

| Notations | Non-Woven Fabric | Solutions |
|---|---|---|
| Blank control | Yes | None |
| Negative control | Yes | 0.9% saline |
| Traucr$^{TM}$ | Yes | 1.0 mg/mL (bovine collagen) |
| THRC-1 | Yes | 0.1 mg/mL (recombinant collagen) |
| THRC-2 | Yes | 0.5 mg/mL (recombinant collagen) |
| THRC-3 | Yes | 1.0 mg/mL (recombinant collagen) |

H&E staining was performed to determine the healing efficacies of THRC-1, THRC-2, and THRC-3 dressings compared with the commercial product (Figure 6B). Immediately after microneedling (0 h), the epidermis was destroyed completely, and the papillary dermis was injured with many inflammatory cells gathering on the wound surfaces (red arrows) in every group, indicating the formation of postoperative defects with acute inflammatory reactions. After two post-treatments at 24 h, the newly formed epidermis (green arrows) in THRC-1, THRC-2, and THRC-3 groups was observed with a clear basal layer, which resembled the newly grown epithelium treated by the commercial product. However, the acute symptoms persisted in the blank control group, and the skin corrosion with 0.9% saline treatment was exacerbated (red arrows). The skin damage was further aggravated in the negative control group at 48 h, while the nascent epidermis with cleavage

was seen in the blank control group, showing delayed wound healing phases of injured tissues than the wounds with bioactive collagen dressings treatments. Notably, the epidermis with full layer structures was generated, and the dermis edema was reduced without cellular infiltration at 48 h in the three THRC dressing groups, as well as the commercial control group, indicating that THRC dressings with different concentrations of THRC (0.1–1.0 mg/mL) and the clinical bovine collagen dressing had similar re-epithelization rates and dermal healing effects. The results showed that THRC dressings with varying amounts of triple-helical recombinant collagen had equivalent healing efficacies to the animal-derived collagen dressing, providing better posttreatment options without allergic reactions.

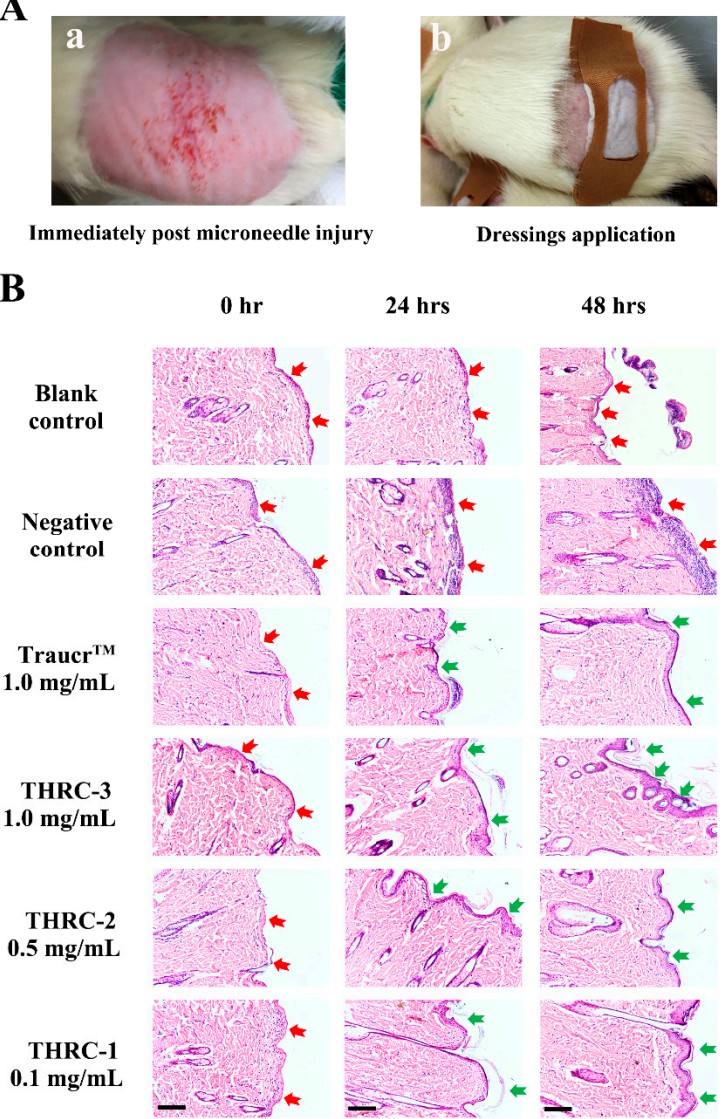

**Figure 6.** The microneedle-injured acute wound healing. (**A**) The experimental process. Immediately after microneedle injury, the epidermis of mice was destroyed with severe erythema (**a**). Subsequently, different dressings were applied with bandage fixation for 20 min (**b**). (**B**) H&E staining images of skin tissues of the blank control, the negative control, and the commercial control Traucr[TM], THRC-3, THRC-2, and THRC-1 groups at different time intervals (0 h, 24 h, and 48 h). The damages of the epidermis were noted by red arrows, while accelerated healing of the epidermis was noted by green arrows. Scale bar: 100 μm.

## 4. Discussion and Conclusions

Microneedling and laser resurfacing procedures have been extensively utilized to rejuvenate facial skin, and they have gained increasing popularity worldwide. However, skin rejuvenation procedures cause adverse skin consequences including persistent pain, erythema, edema, and pigmentation [5,6]. A plethora of dressings such as petrolatum-based ointments and silicone gels have been applied to heal these postoperative skin injuries, while they have reached limited success shadowed by raising concerns of insufficient biocompatibility and unsatisfactory efficacy. Collagen is the vital structural component of the dermis, and collagen dressings have gained popularity in postoperative skin wound care. Compared with the widely used animal-derived collagen, recombinant collagen displays remarkable advantages such as low immunogenicity, no risk of disease transmission, and well-defined quality.

For the first time, we describe the development of biocompatible and nonirritating dressings based on triple-helical recombinant collagen for the accelerated healing of microneedle-injured and photodamaged acute skin wounds. CD analysis of recombinant collagen THRC from different batches demonstrated that THRC consistently maintains a unique triple-helical collagen structure. Cell experiments of the THRC dressings suggest that they possess excellent biocompatibility and bioactivity, pronouncedly promoting the proliferation and adhesion of fibroblasts. The in vivo skin irritation test of New Zealand rabbits further shows that the THRC dressings are very safe and non-irritating.

THRC dressings have been applied to the photodamaged mice wounds, which are rapidly healed without complications in a short period. Photo-injured skin is severely damaged in both the epidermis and the dermis, resulting in acute inflammation and collagen fiber breakage [34]. Skin erythema and edema were observed as acute post-inflammatory reactions, which were significantly reduced after 4 days of treatment with THRC dressings, while the reported wound care products of a petroleum-based ointment or a novel silicone gel showed persistent redness on day 7 [12,37]. Meanwhile, on day 4, H&E-stained tissues revealed only a few cellular infiltrations. Compared to untreated photodamaged defects, THRC dressings improved wound healing by accelerating inflammation. Furthermore, THRC dressings remarkably promoted the arrangement and deposition of collagen fibers, which could be accomplished by supplementing contact collagen, reducing denatured collagen, and stimulating collagen regeneration according to our previous research [38]. It has been reported that keeping an acute wound moist is important for preventing scarring [39]. THRC dressing promotes acute wound healing by providing a moist microenvironment and bioactive collagen. Fibroblasts migrate much more easily through collagen nanofibers from normal sites to wound areas, where they synthesize collagen and growth factors, thereby accelerating tissue regeneration and achieving a satisfactory cosmetic appearance [40].

Various concentrations of THRC dressings showed the same rapid epithelialization rates in rat skin after microneedle injuries as the current clinical dressing, which contains 1.0 mg/mL bovine collagen and has long been used for postoperative wound care. The THRC-1 (0.1 mg/mL THRC), THRC-2 (0.5 mg/mL THRC), and THRC-3 (1.0 mg/mL THRC) dressings all repaired the damaged epidermis and reduced acute dermal inflammation in 48 h, i.e., the same speed as the bovine collagen dressing. THRC is a type of recombinant collagen produced by *E. coli* that would cause much fewer allergic reactions than bovine collagen, which was reported to cause immune responses in 2–4% of the population [41]. Furthermore, THRC dressings with similar bioactivity to animal-derived collagen promote cell proliferation, adhesion, and migration, which speeds up postoperative wound re-epithelialization. Notably, THRC dressings with a low concentration show a similarly fast epithelialization rate as commercial bovine collagen dressings, highlighting the superior efficacy of THRC dressings for curing acute skin wounds.

Recombinant collagen THRC can be produced with high purity at large scales. THRC is highly safe with no virus infection, skin irritation, and cytotoxicity. The biocompatible and bioactive recombinant collagen dressing provides an attractive remedy for postoper-

ative acute skin wounds in facial rejuvenation, which may have great potential in medical cosmetology and dermatology; however, controlled and randomized clinical trials of THRC dressings for postoperative treatments are urgently needed to be conducted.

**Author Contributions:** Data curation, C.F. and H.G.; formal analysis, C.F.; investigation, C.F., S.S., N.W. and Y.F.; methodology, C.F.; supervision, P.L. and J.X.; validation, C.F.; writing—original draft, C.F. and J.X.; writing—review and editing, P.L. and J.X. All authors have read and agreed to the published version of the manuscript.

**Funding:** This work was supported by grants from the National Natural Science Foundation of China (Grant No. 22074057 and 21775059), the Natural Science Foundation of Gansu Province (Grant Nos. 20YF3FA025 and 18YF1NA004), and the Lanzhou Talent Innovation and Entrepreneurship Project (Grant No. 2019-RC-43).

**Institutional Review Board Statement:** All animal experiments were performed with protocols approved by the ethics committee of the College of Chemistry and Chemical Engineering at Lanzhou University (No. G09, 20220711).

**Informed Consent Statement:** Not applicable.

**Data Availability Statement:** Data are available on request from the corresponding author.

**Conflicts of Interest:** The authors declare no conflict of interest. The funders had no role in the design of the study; in the collection, analyses, or interpretation of data; in the writing of the manuscript, or in the decision to publish the results.

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
