# Peer review of "Biocompatible Triple-Helical Recombinant Collagen Dressings for Accelerated Wound Healing in Microneedle-Injured and Photodamaged Skin"

_cosmetics, doi:10.3390/cosmetics10010031_

Round 1
Reviewer 1 Report
The authors describe a new E.coli derived triple helical collagen dressing which could promote and accelerate wound healing, in particular secondary to UV-radiation and microneedling injuries.
They performed in vitro and in vivo experiments to address skin safety, biocompatibility and the efficacy of the product.
The experiment seems well-conducted and clearly presented.
I would suggest to briefly but critically compare in the discussion/conclusion section the efficacy and feasibility of the recombinant collagen of the present study with other commercially available products used to promote wound healing after microneedling (based on literature findings, if relevant), to further enlight the potential benefit (or cons) of THRC dressing.
Table 1 could be modified for a better clearness (more columns dividing the timepoints).
Author Response
Response to Reviewer 1 Comments
Reviewer #1:
“The authors describe a new E.coli derived triple helical collagen dressing which could promote and accelerate wound healing, in particular secondary to UV-radiation and microneedling injuries. They performed in vitro and in vivo experiments to address skin safety, biocompatibility and the efficacy of the product. The experiment seems well-conducted and clearly presented.”
We are pleased that the reviewer thinks our study was well-conducted and clearly presented. We have made specific changes following the reviewer’s advice.
“1.I would suggest to briefly but critically compare in the discussion/conclusion section the efficacy and feasibility of the recombinant collagen of the present study with other commercially available products used to promote wound healing after microneedling (based on literature findings, if relevant), to further enlight the potential benefit (or cons) of THRC dressing.”
We really appreciate the reviewer’s valuable suggestions, and have added more discussions about the comparison of recombinant collagen dressing with other commercial products such as bovine collagen dressing, petroleum-based ointment, and novel silicone gel in the manuscript. “Skin erythema and edema were observed as acute post-inflammatory reactions, which were significantly reduced after 4 days of treatment with THRC dressings, while the reported wound care products of a petroleum-based ointment or a novel silicone gel showed persistent redness on day 7.”
“Various concentrations of THRC dressings show the same rapid epithelialization rates in rat skin after microneedle injuries as the current clinical dressing, which contains 1.0 mg/mL bovine collagen and has long been used for postoperative wound care. The THRC-1 (0.1 mg/mL THRC), THRC-2 (0.5 mg/mL THRC), and THRC-3 (1.0 mg/mL THRC) dressings all repair damaged epidermis and reduce acute dermal inflammation in 48 hours, which is the same speed as the bovine collagen dressing. THRC is a type of recombinant collagen produced by E. Coli that would cause much less allergic reactions than bovine collagen, which was reported to cause immune responses in 2-4% of the population. Furthermore, THRC dressings with similar bioactivity as animal-derived collagen to promote cell proliferation, adhesion, and migration, which speeds up postoperative wound re-epithelialization. Notably, THRC dressings with a low concentration show a similarly fast epithelialization rate as commercial bovine collagen dressings, highlighting the superior efficacy of THRC dressings for curing acute skin wounds.”
“2. Table 1 could be modified for a better clearness (more columns dividing the timepoints).”
We really appreciate the reviewer’s valuable suggestion, and have modified Table 1.
Table 1
Scores of edema and erythema of control (Con) and test (Test) sites of 4 rabbits at various time intervals.
|
1 hr |
24 hrs |
48 hrs |
72 hrs |
||||||||||||
Skin reactions |
Edema |
Erythema |
Edema |
Erythema |
Edema |
Erythema |
Edema |
Erythema |
||||||||
Skin sites |
Con |
Test |
Con |
Test |
Con |
Test |
Con |
Test |
Con |
Test |
Con |
Test |
Con |
Test |
Con |
Test |
Female 1 |
0 |
0 |
0 |
0 |
0 |
0 |
0 |
0 |
0 |
0 |
0 |
0 |
0 |
0 |
0 |
0 |
Female 2 |
0 |
0 |
0 |
0 |
0 |
0 |
0 |
0 |
0 |
0 |
0 |
0 |
0 |
0 |
0 |
0 |
Male 1 |
0 |
0 |
0 |
0 |
0 |
0 |
0 |
0 |
0 |
0 |
0 |
0 |
0 |
0 |
0 |
0 |
Male 2 |
0 |
0 |
0 |
0 |
0 |
0 |
0 |
0 |
0 |
0 |
0 |
0 |
0 |
0 |
0 |
0 |
*Primary Dermal Irritation Index (PDII) = 0/16, PDII = 0.
*Based on the PDII irritation category for THRC dressing (1.0 mg/mL recombinant collagen) is Negligible.

Reviewer 2 Report
This manuscript addresses a study on the biocompatibility, non-irritation, and wound-healing properties of recombinant collagen; this would be relevant and important for helping to improve the effectiveness and safety of wound care treatments. Globally this is a good, well-structured, and scientifically sound manuscript and could contribute to various scientific fields involved in developing and applying wound care products and technologies.
Below I point out some issues that the author should consider to enhance this manuscript:
1. The title is extremely long, and its construction seems more like a sentence (from the abstract or conclusions) than a title. Please revise it.
2. Overall, while the abstract appears to be well-written and provides a clear summary of the main findings and conclusions of the study, it could be further improved by providing more specific information about the methods and results of the cell and animal studies as more context for the study. Also note this: <<Recombinant collagen "THRC"…>> this acronym appears for the first time; it should be explained; Abbreviations should be checked in the whole document to see if they have been spelled out first or explained.
3. The introduction provides a good overview of the current state of the field and the research question and hypothesis. However, it could be further improved by providing more specific information about the current limitations of wound care dressings, the characteristics of the novel recombinant collagen dressing, and more context for the study.
4. At the end of the introduction, something stands out about the melting temperature. The melting temperature of collagen dressings is typically around 60-70°C, which is one reason why collagen dressings are often used in wound care. Any temperature above this would be considered high and could potentially affect the stability and effectiveness of the dressing. Thus, the indicated 34ºC seems too low to me. Can the authors clarify this?
5. The materials and methods sections reasonably describe the study design, sample size and selection, data collection methods, and statistical analyses. However, a potential improvement to this methods section could be to provide more information about the specific rationale for the various procedures.
6. The results section is well structured, and the data is presented with statistical analysis when possible, but there is a lack of graphical quality (maybe due to the pdf format, but please check it).
7. The discussion and conclusions section seems incomplete since it only briefly summarizes the study's main findings and implications, highlighting the potential value of THRC as a treatment for acute skin wounds in facial rejuvenation. Thus, it could be enhanced by providing a more detailed interpretation of the results, comparison to other studies, and discussion of limitations and future directions.
The authors have done good laboratory work. However, the article needs several improvements.
Author Response
We are pleased that the reviewer thought our study is good, well-structured, and scientifically sound. We really appreciate the reviewer’s valuable suggestions and have addressed the concerns one by one.

Reviewer 3 Report
Do you really mean to put the word 'canonic' in the abstract?
Author Response
Thanks for the reviewer’s valuable suggestion. We have removed the word “canonic”, and changed the sentence as follows: “Circular dichroism (CD) measurements of THRC from various batches exhibited triple helical structure characteristic of collagen.”
Round 2
Reviewer 2 Report
The authors addressed the comments properly, the article can be published in the current format.